# CLUSTER-FORMER: CLUSTERING-BASED SPARSE TRANSFORMER FOR QUESTION ANSWERING

## ABSTRACT

Transformer has become ubiquitous in the deep learning field. One of the key ingredients that destined its success is the self-attention mechanism, which allows fully-connected contextual encoding over input tokens. However, despite its effectiveness in modeling short sequences, self-attention suffers when handling inputs with extreme long-range dependencies, as its complexity grows quadratically *w.r.t.* the sequence length. Therefore, long sequences are often encoded by Transformer in chunks using a sliding window. In this paper, we propose *Cluster-Former*, a novel clustering-based sparse Transformer to perform attention across chunked sequences. The proposed framework is pivoted on two unique types of Transformer layer: Sliding-Window Layer and Cluster-Former Layer, which encode local sequence information and global context jointly and iteratively. This new design allows information integration beyond local windows, which is especially beneficial for question answering (QA) tasks that rely on long-range dependencies. Experiments show that Cluster-Former achieves state-of-the-art performance on several major QA benchmarks.

## 1 INTRODUCTION

Long-range contextual understanding has proven critical in many natural language processing (NLP) tasks. For example, the relevant context for correctly answering an open-domain question can arch over thousands of words. Encoding long sequences via deep neural networks, however, has remained an expensive and challenging task due to high demand on training time and GPU memory. Traditional sequence modeling methods (Hochreiter & Schmidhuber, 1997) encode long sequences in a chronological order, which suffers high latency. In the place of sequential encoding, recent models such as Transformer (Vaswani et al., 2017) use simultaneous self-attention over the entire input instead, which has been successfully adopted in many NLP tasks such as textual entailment (Devlin et al., 2019), dependency parsing (Zhou & Zhao, 2019), and summarization (Lewis et al., 2019). A caveat with Transformer though is that building full connections over long sequences translates to quadratic growth on memory demand and computational complexity *w.r.t.* sequence length.

One way to efficiently encode long sequences is to first chunk a sequence into much shorter ones with a sliding window, then build connections between the shorter sequences (Figure 1(a)). For example, Child et al. (2019), Beltagy et al. (2020) and Zaheer et al. (2020) apply sparse attention to chunked sequences in hand-designed patterns in order to gather information from the chunks (Figure 1(b)). Choi et al. (2017) and Wang et al. (2019) first use a simpler model to filter chunked sequences, then process selected sequences with fully-connected self-attention. Rae et al. (2019) makes use of the shared memory of chunked sequences to build connections between them. However, these methods cannot encode long-range dependencies with as much flexibility or accuracy as fully-connected self-attention, due to their dependency on hand-designed patterns.

Recently, several studies (Kitaev et al., 2020; Tay et al., 2020b) propose to further improve the sparse attention mechanism by hashing or sorting the hidden states into different buckets (Figure 1(c)). These works mainly explore tasks with relatively short sequences, such as sentence-level Machine Translation (MT), where the number of hashing vectors is relatively small (less than 16 in Kitaev et al. (2020)), allowing randomly initialized hashing vectors to hash hidden states into correct buckets. However, how to use hashing-based attention in the context of long sequences (*e.g.,*, up to thousands of words) is still an unexplored territory.

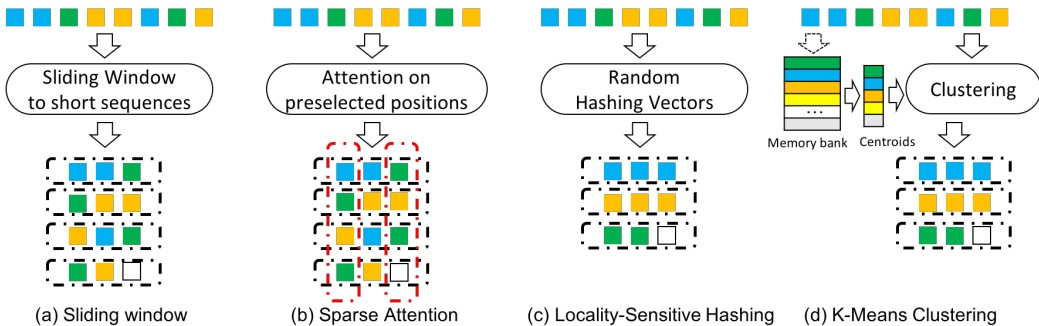

Figure 1: Illustration of different methods for processing long sequences. Each square represents a hidden state. The black-dotted boxes are Transformer layers. (a) is the sliding-window-based method to chunk a long sequence into short ones with window size 3 and stride 2. (b) builds cross-sequence attention based on sliding window over pre-selected positions (red-dotted boxes). (c) hashes the hidden states into different buckets by randomly-initialized vectors. (d) is our proposed approach to cluster the hidden states. Our final model is a combination of (a) and (d) that processes both local and global context.

Our proposed framework for efficient long sequence encoding, *Cluster-Former*, marries both sliding-window and hashing-based methods to achieve effective local and long-range dependency encoding. Cluster-Former consists of two types of encoding layer. The first one (noted as *Sliding-Window Layer*) focuses on extracting local information within a sliding window. It applies Transformer to the hidden states of each chunked sequence independently, as shown in Figure 1(a). The other one (noted as *Cluster-Former Layer*) learns to encode global information beyond the initial chunked sequences. Specifically, we first apply clustering to the input hidden states so that similar hidden states are assigned to the same cluster, as shown in Figure 1(d). The clustered and sorted input is then divided uniformly into chunks, each encoded by a Transformer layer. Note that to make model training more efficient, the cluster centroids are not computed online but updated periodically (every epoch or a few epochs). We accumulate the hidden states from the layer prior to the Cluster-Former layer in a memory bank, and apply the K-Means algorithm to form cluster centroids during each update cycle. Compared to previously discussed sparse attention based on pre-selected positions (Figure 1(b)) or randomly-initialized hashing vectors (Figure 1(c)), experimental results show that our method can encode dependency across chunked sequences more effectively.

Our contributions can be summarized as follows. ($i$) We propose Cluster-Former, a novel approach to capturing long-range dependencies more effectively than locality-sensitive hashing method. ($ii$) We propose a new Transformer-based framework to process long sequences by combining Sliding-Window and Cluster-Former layers to extract both local and global contextual information. ($iii$) Our model achieves the best performance on question answering datasets of Natural Questions (long answer), SearchQA, and Quasar-T.

## 2 RELATED WORK

**Efficient Transformers** With Transformer models growing larger and larger, how to handle longer sequences arises as a critical challenge. Many works have been proposed to improve the computational and memory efficiency of Transformers, including Sparse Transformer (Child et al., 2019), Routing Transformer (Roy et al., 2020), Reformer (Kitaev et al., 2020), Sinkhorn Transformer (Tay et al., 2020b), Longformer (Beltagy et al., 2020), ETC (Ainslie et al., 2020), Synthesizer (Tay et al., 2020a), Performer (Choromanski et al., 2020), Linformer (Wang et al., 2020), Linear Transformer (Katharopoulos et al., 2020), and BigBird (Zaheer et al., 2020). Tay et al. (2020c) provided an excellent literature survey on this emerging topic. Our method falls into the setting of learnable sparse-attention patterns including Routing Transformer, Reformer and Sinkhorn Transformer. Our method is closer to Routing Transformer (Roy et al., 2020) which also uses cluster centroids to learn patterns, while we are targeting on quite different tasks (language modeling VS question answering) which leads to the significant difference on frameworks. Moreover, our cluster centroids are updated in very different ways (online exponentially moving centroids VS periodical centroids update by KMeans).

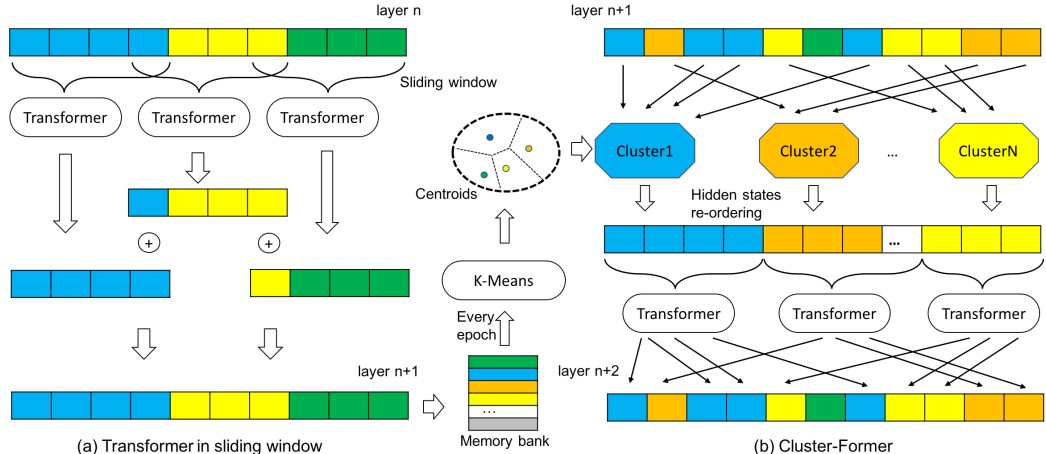

Figure 2: An overview of proposed Transformer layer. (a) Sliding-Window layer over a sequence. (b) Cluster-Former layer over clustered hidden states from the output of (a). Cluster centroids are periodically updated based on the memory bank of the hidden states in the corresponding layer. Note that the sequence inputs in (a) and (b) usually come from two different samples.

**Long Sequence in Question Answering** For tasks such as open-domain question answering (Chen et al., 2017), a large volume of documents or paragraphs is usually retrieved to infer the answer, yielding extremely long context content. Despite the fact that state-of-the-art NLP models are capable of extracting answers amid complex context, they still struggle with extremely long input sequences. Recent advances that advocate the use of large-scale pre-trained models (Lewis et al., 2019; Liu et al., 2019; Lan et al., 2020) for question answering make this problem more prominent, due to tremendous memory consumption. To process long sequences, the most widely-used method is to first use a lightweight model to filter out redundant text, then use sliding-window-based approaches to encode the remaining sequences with a more sophisticated model. Chen et al. (2017) integrated bi-gram features into Information Retrieval (IR) methods to retrieve related documents more accurately. Wang et al. (2018) trained a paragraph selector using as the reward whether the entire system can obtain the correct answer or not . Lin et al. (2018) proposed to use a paragraph ranking model to curate data that are required for training reading comprehension models. Wang et al. (2019) trained a ranker to merge paragraphs for multi-passage reasoning. Asai et al. (2020) trained a recurrent retriever to select paragraphs for multi-hop question answering. Besides the above methods, directly applying Efficient Transformers to process long sequences in question answering is another option. In this paper, we focus on this direction by directly training our Cluster-Former on the long context without using lightweight model for context filtering.

## 3 PROPOSED APPROACH

The proposed framework to handle long sequences is pivoted on two types of Transformer layer: ($i$) *Sliding-Window* Layer; and ($ii$) *Cluster-Former* Layer. The former focuses on encoding local sequence information, while the latter is on encoding global context and always built on top of the former layer. An overview of the two layers is illustrated in Figure 2.

### 3.1 SLIDING-WINDOW LAYER

Despite that our focus is on capturing long-range dependencies for global context, local information also plays a critical role for knowledge propagation. Therefore, in the lower section of our network, we adopt the traditional sliding-window encoding mechanism. A sliding window segments a long sequence $X$ into short, overlapping ones with window size $l$ and stride $m$, as illustrated in Figure 2(a). Note that in this paper, we focus on question answering tasks, for which we concatenate the question $Q$ with each sequence chunked from the document:

$$\mathbf{H}_k^0 = [\mathbf{Q}; \mathbf{X}\left[m \times k : (m \times k + l)\right]], \tag{1}$$

where $\mathbf{Q} \in \mathbb{R}^{q \times d}$ denotes question embeddings given a QA task, $q$ is the number of tokens in the question, and $\mathbf{X} \in \mathbb{R}^{x \times d}$ is the embeddings for all context, $x$ is the number of tokens in context. $k$ is the ID of the chunked sequence, $l$ is the window size, and $m$ is the stride of the sliding window. $[idx_1 : idx_2]$ indicates selecting rows between the index of $idx_1$ and $idx_2$ of the matrix. $[\cdot ; \cdot]$ means concatenating the matrices along the row. We first use Transformer to encode each sequence in sliding window as follows:

$$\mathbf{H}_k^{n+1} = \text{Transformer}(\mathbf{H}_k^n), \tag{2}$$

where $\mathbf{H}_k^{n+1} \in \mathbb{R}^{(q+l) \times d}$ is the output of Transformer on the $k$-th sequence in the $n$-th layer. While it is not the final output of the $n$-th layer. As we expect the neighbouring sequences to share useful information in hidden states as well, we always set $m < l$ to allow overlapping between sequences. We use the mean values of the Transformer hidden states at the overlapped tokens between windows as final outputs. To merge the representations from $(k-1)$-th sequence:

$$\mathbf{H}_k^{n+1}[q : q+l-m] \quad += \quad \mathbf{H}_{k-1}^{n+1}[q+m : end],$$
$$\mathbf{H}_k^{n+1}[q : q+l-m] \quad /= \quad 2,$$

and merge representations from $(k+1)$-th sequence:

$$\mathbf{H}_k^{n+1}[q+m : end] \quad += \quad \mathbf{H}_{k+1}^{n+1}[q : q+l-m],$$
$$\mathbf{H}_k^{n+1}[q+m : end] \quad /= \quad 2, \tag{3}$$

where $+=$ is to add matrices in-place and $/=$ is to divide a matrix by a scalar value in-place. The merged hidden states $\mathbf{H}_k^{n+1} \in \mathbb{R}^{(q+l) \times d}$ are the final outputs of the $n$-th layer. If the next layer is Cluster-Former, the output hidden states in this layer $\mathbf{H}_k^{n+1}$ will be saved into memory bank for computing the cluster centroids.

## 3.2 CLUSTER-FORMER LAYER

We introduce a Cluster-Former layer to add global representational power to Transformer beyond sliding windows. An in-depth visualization of the layer is illustrated in Figure 2(b).

The input of the Cluster-Former layer comes from the hidden states of the prior layer (in our case a Sliding-Window layer). After merging the overlaps between sequence chunks, the input of this layer is defined as:

$$\bar{\mathbf{H}}^n = [\mathbf{H}_0^n[0 : q+m]; ...; \mathbf{H}_k^n[0 : q+m]], \tag{4}$$

where $\bar{\mathbf{H}}^n \in \mathbb{R}^{(q\lceil x/m \rceil + x) \times d}$ is the hidden states to cluster, $x$ is the number of tokens in the context.

As the hidden states with larger cosine similarity are more likely to have higher attention weights, we build sparse self-attention only on the hidden states in the same cluster. In this work, we use K-Means as the chosen clustering method for simplicity. More advanced clustering algorithms have the potential of yielding better performance. Since running K-Means on the fly in each training iteration is computationally expensive, we decide to re-compute the cluster centroids with low frequency (every epoch or a few epochs).

**Algorithm 1** Cluster Centroids Update

```
1: Initialize Memory = Queue()
2: Centroids = GETCENTROIDS(RandomVector)
3:
4: function TRAIN(Inputs)
5:     for i = 1, 2,…, IterationNum do
6:         States = Sliding-Transformer(Inputs[i])
7:         Memory.add(States)
8:         while len(Memory) > M do
9:             Memory.pop()
10:        end while
11:        if i % ClusterUpdateFrequency == 0 then
12:            Centroids = GETCENTROIDS(Memory)
13:        end if
14:        Clusters = cluster States by Centroids
15:        States = Cluster-Former(Clusters)
16:    end for
17: end function
18:
19: function GETCENTROIDS(HiddenStates)
20:     Centroids = K-Means(HiddenStates)
21:     Outputs = List()
22:     Outputs[1] = Centroids[1]
23:     for i = 2, 3,…, ClusterNum do
            Outputs[i] = centroid from Centroids
24:                     that is closest to Outputs[i − 1]
                        but not in Outputs
25:     end for
26:     return Outputs
27: end function
```

In addition, to avoid dramatic changes in the cluster centroids due to limited hidden state inputs, we maintain a memory bank for the most recent hidden states. The entire procedure is depicted in Algorithm 1. Once we compute the cluster centroids, we can directly use them for hidden state clustering as follows:

$$\mathbf{v}^n = \operatorname{argmax}\Big(\frac{\mathbf{H}^n(\mathbf{C}^n)^T}{||\mathbf{H}^n||_2||\mathbf{C}^n||_2}\Big), \tag{5}$$

where $\mathbf{C}^n \in \mathbb{R}^{p \times d}$ are the cluster centroids for layer $n$, and $p$ is the pre-defined number of clusters. The function $\operatorname{argmax}(\cdot)$ performs on the last dimension and assigns all the input hidden states into different clusters based on the max value of cosine similarity between the hidden states and cluster centroids. $\mathbf{v}^n \in \mathbb{R}^{(q\lceil x/m \rceil + x)}$ is the assigned cluster IDs of all the input hidden states.

Since the number of hidden states in different clusters can vary substantially, padding them to the maximum length for Transformer training will significantly increase the computational time. To make the extraction of global context more efficient, we greedily pick the cluster centroids based on the nearest neighbour (measured by cosine similarity) as shown in the function GETCENTROIDS in Algorithm 1. Thus, the hidden states with similar cluster IDs are also close to each other. Then, we can directly sort the cluster IDs of hidden states and uniformly chunk the hidden states (same window size and stride $m$):

$$\mathbf{u}^n = \operatorname{argsort}(\mathbf{v}^n), \quad \mathbf{a}_k^n = \mathbf{u}^n[mk : m(k+1)], \quad \mathbf{E}_k^n = \mathbf{H}^n[\mathbf{a}_k^n], \tag{6}$$

where the function $\operatorname{argsort}(\cdot)$ is to obtain the indexes of input values sorted in order (same values sorted by the corresponding position of hidden states). $\mathbf{a}_k^n \in \mathbb{R}^m$ is the chunked indexes of the hidden states. $\mathbf{E}_k^n \in \mathbb{R}^{m \times d}$ is the $k$-th clustered hidden states, and we will run Transformer on top of it to build the connection beyond the words in the initial sliding window as follows:

$$\mathbf{E}_k^{n+1} = \operatorname{Transformer}(\mathbf{E}_k^n). \tag{7}$$

After updating the hidden states, we map them back to the order before clustering:

$$
\begin{aligned}
\bar{\mathbf{H}}^{n+1} &= [\mathbf{E}_0^{n+1}; \mathbf{E}_1^{n+1}; ...; \mathbf{E}_K^{n+1}], \\
\bar{\mathbf{a}}^n &= [\mathbf{a}_0^n; \mathbf{a}_1^n; ...; \mathbf{a}_K^n], \tag{8}\\
\bar{\mathbf{H}}^{n+1}[\bar{\mathbf{a}}^n] &= \operatorname{clone}(\bar{\mathbf{H}}^{n+1}), \tag{9}
\end{aligned}
$$

where $\bar{\mathbf{H}}^{n+1}$ is the final output hidden state of this layer and has the same word order as the input $\bar{\mathbf{H}}^n$. In experiments, we stack these two types of layer interchangeably to capture both global and local context efficiently.

## 4 EXPERIMENTS

### 4.1 DATASETS

We evaluate our proposed approach on multiple question answering benchmarks. The statistics of all the datasets are summarized in Table 1.

**Quasar-T**[1] (Dhingra et al., 2017): The goal of this task is to answer open-domain questions from Trivia Challenge. All the passages harvested through information retrieval can be used to answer questions. The task requires the model to generate answers in phrases. The evaluation metric on this dataset is based on Exact Match and F1 score of the bag-of-words matching. Our evaluation tool[2] comes from the SQuAD dataset.

**SearchQA**[3] (Dunn et al., 2017): The setting of this dataset is the same as Quasar-T, except that the questions are sourced from Jeopardy! instead.

|  | #train | #test | med | max |
|---|---|---|---|---|
| Quasar-T | 29k | 3k | 2.8k | 8.2k |
| SearchQA | 100k | 27k | 2.5k | 4.9k |
| NQ | 292k | 8k | 6.3k | 128k |

Table 1: Statistics of Question Answering datasets. #train: number of questions in the training set. #test: number of questions in the test set. med: median length of the context. max: max length of the context.

---

[1] https://github.com/bdhingra/quasar
[2] https://rajpurkar.github.io/SQuAD-explorer/
[3] https://github.com/nyu-dl/dl4ir-searchQA

**Natural Questions**[4] (Kwiatkowski et al., 2019): This task aims to answer questions based on a given Wikipedia document, and has two settings. ($i$) Long answer: select a paragraph that can answer the question based on the Wikipedia document if any. ($ii$) Short answer: extract an answer phrase from the document if the document contains the answer. As the given document may not contain answer, we can either predict an answer or predict no answer. The evaluation metric on this dataset is the F1 score, where true positives are exactly correct answers, false positives are incorrect answer predictions, and false negatives are incorrect "no answer" predictions. As the test set is hidden, we split 5% of the training set for validation, and use the original validation set for testing. We use the official tool from the dataset to evaluate our models. We also submit our best model to the leaderboard.

## 4.2 IMPLEMENTATION DETAILS

All the models are trained on 8 Nvidia V100 GPUs. For clustering, we adopt "Yinyang kmeans "(Ding et al., 2015)[5] which takes less than 5 seconds for clustering in all of our experiment settings. We set the memory size for clustering $M = 100,000$ in Algorithm 1. We use cluster centroids that perform the best on the validation set for test set experiments. We initialize our models with RoBERTa-large (Liu et al., 2019). As the number of position embeddings of RoBERTa is limited to 512, we cannot assign different position embeddings to all tokens. Instead, we assign the same position embeddings to each chunked sequence. The majority of our model is made up of Sliding-Window Layers, as the local information is essential for QA tasks. We adopt the proposed Cluster-Former Layer in layers 15 and 20 to further capture long-range information. We set the sliding window size $l$ to 256, stride $m$ to 224, and change the number of clusters in $\{64, 256, 512\}$ to analyze its impact on the final performance. We prepend a special token to the beginning of all the given/retrieved paragraphs and directly concatenate all the paragraphs as the final context sequence. Due to memory constraints, we set the max length to be 5000 during training and 10000 during inference. During dataset finetuning, we use Adam (Kingma & Ba, 2015) to optimize the model. We set warm-up updates to 2,220, maximal updates to 22,200, learning rate to $5 \times 10^{-5}$, and batch size to 160. We tune dropout rate from $\{0.1, 0.15, 0.2\}$ for all methonds including baselines and report the best results. The model converges in one day for all the QA datasets.

For Quasar-T and SearchQA, we predict the start and end positions of the answer. For Natural Question, we first identify whether the question has short/long answers or not based on the mean values of the first hidden state of all the chunked sequences, $\frac{1}{K} \sum_{k=1}^{K} \mathbf{H}_k^N[0]$ , where $K$ is the number of chunks and $N$ is the number of layers. If answerable, we rank all the candidates for long answer selection, and predict the start and end positions of short answers. Our model submitted to Natural Question Leaderboard ensembled 3 models with 512 clusters, and only these models are firstly trained on SQuAD2.0 and then finetuned on Natural Question dataset.

## 4.3 BASELINE

We compare our models with several strong baselines, including:

**R3** (Wang et al., 2018) proposes to use reinforcement learning to jointly train passage ranker and reader. **DS-QA** (Lin et al., 2018) proposes to first use paragraph selection to filter the noisy data and then trained model on denoised data. **Multi-passage BERT** (Wang et al., 2019) proposes to filter the passages and then merge multiple useful passages into one sequence, which can be encoded by BERT. **DrQA** (Chen et al., 2017) makes use of attention mechanism across the question and the document for answer phrase extraction. **DecAtt and DocReader** (Kwiatkowski et al., 2019) is based on a pipeline approach that first uses a simpler model to select long answers and then a reading comprehension model to extract short answers from the long answers. **BERT$_{joint}$** (Alberti et al., 2019) jointly trains short and long answer extraction in a single model rather than using a pipeline approach. **BERT$_{wwm}$+SQuAD2** (Pan et al., 2019) makes use of multi-task learning to further boost performance. **RikiNet-RoBERTa** (Liu et al., 2020) proposes a dynamic paragraph dual-attention reader and a multi-level cascaded answer predictor. **BigBird-ETC** (Zaheer et al., 2020) makes use of a sparse attention mechanism to encode long sequences.

---

[4]https://ai.google.com/research/NaturalQuestions
[5]https://github.com/src-d/kmcuda

|  | Quasar-T EM/F1 | SearchQA EM/F1 | NQ(long) F1 | NQ(short) F1 |
|---|---|---|---|---|
| R3 (Wang et al., 2018) | 35.3/41.7 | 49.0/55.3 | - | - |
| DECAPROP (Tay et al., 2018) | 38.6/46.9 | 62.2/70.8 | - | - |
| DS-QA (Lin et al., 2018) | 42.2/49.3 | 58.8/64.5 | - | - |
| Multi-passage BERT (Wang et al., 2019) | 51.1/59.1 | 65.1/70.7 | - | - |
| DrQA (Chen et al., 2017) | 37.7/44.5 | 41.9/48.7 | 46.1 | 35.7 |
| DecAtt + DocReader (Kwiatkowski et al., 2019) | - | - | 54.8 | 31.4 |
| BERT$_{joint}$ (Alberti et al., 2019) | - | - | 64.7 | 52.7 |
| BERT$_{wwm}$ + SQuAD2 (Pan et al., 2019) | - | - | 68.2 | 57.2 |
| RikiNet-RoBERTa (Liu et al., 2020) | - | - | 75.3 | **59.3** |
| Sliding Window | 52.9/62.8 | 65.8/73.2 | 75.3 | 56.4 |
| Sparse Attention (Child et al., 2019) | 52.1/62.0 | 64.7/71.7 | 74.5 | 56.1 |
| Locality-Sensitive Hashing (Kitaev et al., 2020) | 53.2/62.9 | 66.0/73.5 | 75.5 | 56.4 |
| Cluster-Former (#C=64) | 53.3/63.3 | 67.0/74.2 | 76.3 | 56.7 |
| Cluster-Former (#C=256) | 53.6/63.5 | 67.5/74.5 | 76.3 | 56.7 |
| Cluster-Former (#C=512) | **54.0/63.9** | **68.0/75.1** | **76.5** | 57.1 |

Table 2: Results on Quasar-T, SearchQA test sets and NQ dev set. #C: number of clusters.

|  | Long Answer | | | Short Answer | | |
|---|---|---|---|---|---|---|
|  | F1 | Precision | Recall | F1 | Precision | Recall |
| BigBird-ETC-large (Zaheer et al., 2020) | 77.8 | 77.5 | **78.1** | 57.9 | 63.7 | 53.0 |
| RikiNet (Liu et al., 2020) | 76.1 | 78.1 | 74.2 | **61.3** | **67.6** | 56.1 |
| Cluster-Former (Ours) | **78.0** | **78.5** | 77.5 | 60.9 | 62.1 | **59.8** |

Table 3: Results on Natural Questions (NQ) leaderboard (test set). We show two published results here from over 40 submissions. And our model achieves No.1 for long answer and No.4 for short answer.

We also re-implement several strong baselines which have not been applied to process long context in question answering tasks:

- **Sliding Window**: The original method is fully made up of Sliding-Window Layers and can only attend to local information.

  To make a fair comparison among different methods on long-range information collection, we replace several layers of this sliding window baseline with Sparse Attention, Locality-Sensitive Hashing, and Cluster-Former.

- **Sparse Attention** (Child et al., 2019): This method replaces several layers in the previous baseline by training a Transformer layer across sequences on pre-selected positions. We run this sparse Transformer on all the hidden states in the same position across sequences, so that the output of sparse Transformer can merge the information from different sequences.

- **Locality-Sensitive Hashing** (Kitaev et al., 2020): This method hashes hidden states into different buckets determined by randomly-initialized hashing vectors. A Transformer layer is then applied across buckets to build Sparse Attention across the whole sequence. Note that this method cannot be directly used for question answering without adding Sliding-Window layer, as our QA model is initialized by RoBERTa that only has 512 position embeddings.

### 4.4 EXPERIMENTAL RESULTS

**State-of-the-Art Results on QA** Table 2 and 3 show that our proposed method outperforms several strong baselines, thanks to its ability to encode both local and global information. Cluster-Former with 512 clusters achieves new state-of-the-art results on Quasar-T, SearchQA and Natural Question (long answer).

**Effect of Cluster-Former** We also test the ability of Cluster-Former on modeling long-range dependencies. Note that Sparse Attention (Child et al., 2019) and Locality-Sensitive Hashing (Kitaev et al., 2020) have never been tested on question answering tasks with long con-

| Question | Where did the underground railroad start and finish ? |
|----------|--------|
| Context | The Underground Railroad by artist Charles T. Webber , 1893 Date Late 1700s - 1865 Location Northern United States with routes to Canada , Mexico ... |
| Special token | Island island in the colonies cityWith in the in . |
| Time | did start and finish 1893 Date 1700 1865 Location Participants Outcome Deaths 19 1763 |
| Stopwords | the the , the , , , , to , , , , the American runaway slaves of free states the , , , it to , a the the |
| Entity | Canada Mexico Canada is applied Florida Spanish Railroad Railroad Railroad |
| Positions | 49, 50, 51, 52, 53, 54, 55, 115, 116, 168, 273, 394, ..., 6022, 6040, 6042, 6060, 6094, 6095 |

Table 6: An example from Natural Question dataset. The rows in the middle section show the corresponding words of the clustered hidden states, and the bottom row shows the positions of the clustered hidden states. "" refers to start token of long answer candidate.

text. For fair comparison, we set the layers 15 and 20 as either Sparse Attention, Locality-Sensitive Hashing or our Cluster-Former, and the left layers are Sliding Window layers.

As shown, Sparse Attention performs worse than our Cluster-Former. The loss may come from the noise introduced by pre-selected positions, the corresponding words of which may not be related. We set the number of hashing vectors in Locality-Sensitive Hashing (LSH) to 64, the same as the number of clusters in Cluster-Former. LSH outperforms the baseline slightly on QA and consistently underperforms our Cluster-Former (#C=64). Overall, our Cluster-Former performs the best.

**Effect of Number of Cluster Centroids** We also test the effect of different numbers of cluster centroids ($C$) on model performance. We observe that the model with 512 clusters works significantly better than the model with 64 clusters on most of the tasks. However, for Natural Questions Long Answer setting, the improvement is marginal. As we mainly rely on the hidden state of special tokens "" for long answer selection, and the same tokens can be assigned into same chunk more easily even with a smaller number of clusters.

|    | 3 | 4 | 5 | 6 |
|----|------|------|------|------|
| 8  | 55.7/65.0 | 55.6/64.4 | 54.7/64.3 | 55.4/64.6 |
| 12 | 55.1/64.9 | 55.8/65.0 | **56.1/65.4** | 55.4/64.6 |
| 16 | 55.6/65.0 | 55.2/64.7 | 55.1/64.6 | 54.8/64.1 |
| 20 | 54.8/64.2 | 55.4/64.8 | 55.1/64.6 | - |

Table 4: Experiments on Quasar-T dev dataset. $a \in \{3, 4, 5, 6\}$ and $b \in \{8, 12, 16, 20\}$, if the layer number $l \% a == 0$ and $l >= b$, we set it as Cluster-Former Layer, otherwise Sliding Window Layer.

|    | Wikitext ppl | Enwik8 bpc |
|----|------|------|
| Sliding window | 20.8 | 1.34 |
| Sparse Attention | 20.5 | 1.29 |
| Locality-Sensitive Hashing | 20.8 | 1.33 |
| Cluster-Former (#C=64) | 20.5 | 1.28 |
| Cluster-Former (#C=256) | 20.3 | 1.24 |
| Cluster-Former (#C=512) | **20.2** | **1.22** |

Table 5: Results on Language Modeling. #C: number of clusters; Wikitext: Wikitext-103.

**Selection of Cluster-Former Layers** We also have an analysis on which layers are better used for Cluster-Former layer. As shown in Table 4, we conduct a hyper-parameter search. And find that it can get better performance with at least one Cluster-Former layers in the middle layer (8-16). The worst results come from only one Cluster-Former layer in the layer of 22 or 23.

**Language Modeling** Although we focus on QA tasks in this paper, to demonstrate the versatility of Cluster-Former, we further conduct additional experiments on language modeling using the Wikitext-103 (Merity et al., 2017) and Enwik8 (Mahoney, 2011) benchmarks. Implementation details are provided in Appendix. As shown in Table 5, Cluster-Former outperforms strong state-of-the-art baselines.

## 4.5 QUALITATIVE ANALYSIS

We perform qualitative analysis on how the hidden states are clustered, by visualizing the corresponding words and positions of the hidden states in Table 6. From the first row, we can see that the special tokens "" tend to belong to the same cluster. Note that "" is the start token of each long answer candidate, and its hidden state is used for final long answer selection. Therefore, Transformer on this cluster can compare across the candidates to make the final prediction.

We further observe that the same types of token are more likely to appear in the same cluster. For example, words from the second row to the forth row cover the topics of time, stopwords, and organization & geopolitical entities.

Finally, we randomly sample a cluster and list the positions of clustered hidden states in the last row of the table. We find that states in long distance, such as the 50-th and 6060-th states (over 6000 tokens apart), can be in one cluster, which demonstrates the ability of Cluster-Former in detecting long-range dependencies. Further, we observe that states tend to cluster in phrases. For example, we see consecutive positions such as "49, 50, 51, 52, 53, 54, 55", which likely results from the sliding-window encoding.

## 5  CONCLUSION

In this paper, we present Cluster-Former, a new method to encode global information for long sequences. We achieve new state of the art on three question answering datasets: Quasar-T, SearchQA, and Natural Questions. Further, we observe that a larger number of clusters in Cluster-Former can lead to better performance on question answering tasks. Cluster-Former is a generic approach, and we believe that it can benefit other NLP tasks that rely on long-range dependencies as well.

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
