# OpenReview forum: "Cluster-Former: Clustering-based Sparse Transformer for Question Answering"
_ICLR.cc/2021/Conference — Reject_

### Official Review · AnonReviewer2 · 2020-10-27
**Strong empirical results for efficient transformer model, questions about related work & analysis.**

**Rating:** 6
**Confidence:** 3

**Review:**

**Summary:**

This paper introduces the ClusterFormer, a transformer architecture that scales gracefully to long sequences by restricting pairwise attention by the cluster assignments of the hidden states of input tokens. The paper presents strong empirical results on question answering benchmark datasets outperform state-of-the-art approaches as well as strong baselines introduced by the authors.

Summary of review: Strong empirical results on question answering datasets; interesting data-driven efficient transformer model; further clarification on relationship to related work needed; experimental results would be stronger with more analysis of the proposed method.

**Strengths:**

The all pairs self attention component of transformers limits their scalability to long sequences. This paper presents a model that is reduces the complexity by grouping related tokens into clusters, such that self-attention is applied only within each cluster. In particular, a long sequence is first encoded using a sliding window style approach, then these sliding window representations are clustered and the resulting cluster memberships determine the sparsity for the remaining layers of the transformer. The approach appears to work quite well on question answering datasets for which the approach achieves state-of-the-art results on three datasets.

The paper is well written and the presentation is very clear.


**Weaknesses:**

**Relationship to related work:** The proposed approach appears to share many similarities to the Routing Transformer (Roy et al, 2020). While both approaches from this year, I think that it would be important to present the similarities and differences of the two approaches (i.e. sliding windows, way k-means centers are updated, etc) clearly in this paper. Other related, though more distinct, ideas are used in the inducing point based variant of Set Transformers (Lee et al, 2019).

**Empirical Analysis of Scaling to Long Sequences:** I think the presentation of the paper would be improved if the authors demonstrated just how much computation is saved by using these sparse, cluster-based attention layers. It would also improve the presentation to compare the efficiency of the proposed approach to other methods at varying input length sizes. Similarly, it would be interesting to show the performance of the proposed approach compared to baselines for varying maximum sequence lengths. It would further be interesting to investigate the cluster centers discovered by the method, what they represent, and how they change over time. This would be particularly important to analyze how the model picks up information across long sequences (i.e., showing that clusters are not made up of tokens from the same sliding window).

**Details of k-means**: Apologies if I've missed this, but is anything done to ensure that the cluster sizes produced by k-means are relatively balanced? The skew of these sizes will directly impact the scalability of the method? Further, while it is implied by the method/text, it would be nice to describe how the gradient is calculated for this hard cluster assignment.


Aurko Roy, Mohammad Saffar, Ashish Vaswani, David Grangier. Efficient Content-Based Sparse Attention with Routing Transformers. First posted March 2020. https://arxiv.org/abs/2003.05997

Juho Lee, Yoonho Lee, Jungtaek Kim, Adam R. Kosiorek, Seungjin Choi, Yee Whye The. Set Transformer: A Framework for Attention-based Permutation-Invariant Neural Networks. ICML 2019.
http://proceedings.mlr.press/v97/lee19d/lee19d.pdf

**Questions for the authors:**

• Please see questions in the details of k-means section.

---

> ### Author Response · Authors · 2020-11-19
> **Response to Reviewer #2**
>
> Thank you for your encouraging and insightful comments. We have updated the draft with the modifications in blue. Below, we provide detailed responses to your questions.
>
> Q1: Relationship to related work.
>
> A1: Thanks a lot for your valuable suggestions! For the comparison to Routing Transformer, please refer to the answer for Reviewer 1. And for Set Transformers, it is more like a memory and low-rank based method by projecting the key and value vectors from attention mechanism into low dimension space. We also think it is different from our work. Besides, one major problem of these frameworks is that they cannot fully make use of existing pretraining models, such as BERT, RoBERTa, ALBERT. While our model is more general and can be easily initialized by general Transformer frameworks to target on SOTA of different tasks. We have added further analysis of the related works in the updated draft.
>
> Q2: Empirical Analysis of Scaling to Long Sequences.
>
> A2:
> 1) As the classic Transformer cannot encode long sequences with 5K tokens, we cannot make a fair empirical comparison of encoding time. And we would like to say sliding windows is a necessary step to encode long sequences by reducing complexity from O($n^2$) to O(nl), where n is the sequence length and l is the window size. The complexity of our method is also O(nl).
> 2) For the experiments of running on varying input length sizes, as we focus more on the question answering tasks and the answer may not appear in the context with short sequence. It is difficult to say whether the performance changes come from lower answer recall or the ability of long dependency detection. The more context we have, the better performance we can achieve for OpenQA (like Quasar-T and SearchQA) has been proved by other works, such as Leveraging Passage Retrieval with Generative Models for Open Domain Question Answering.
> 3) For the clustering details, as shown in the last row of Table 6, we find that states in long distance, such as the 50-th and 6060-th states (over 6000 tokens apart), can be in one cluster, which demonstrates the ability of Cluster-Former in detecting long-range dependencies.
>
> Q3: Details of k-means
>
> A3: Our K-Means is more like first sorting the hidden states and then chunking them, so that it will not have the unbalance issue. As shown in Algorithm 1, we also greedily sort the cluster centers, so that the hidden states with closer cluster ids are also similar to each other.

---

### Official Review · AnonReviewer3 · 2020-10-28

**Rating:** 5
**Confidence:** 3

**Review:**

The paper describes a method to handle long documents for question answering. Most existing approaches use a sliding window approach, without communication between different sliding windows. Instead, they propose an approach that clusters individual vectors, and allows communication (attention) among the locations in the same cluster. I am not sure about the intuition behind this -- why would communicate between similar vectors more efficient than dissimilar or randomly chosen vectors? Would the performance improve if you use a better clustering algorithm? The authors do not provide much intuition on this either.

I have a concern about comparison with locality sensitive hashing. The number of buckets used in locality sensitive hashing was 64. And it's clear that having more clusters help. And the comparison between #C=64 Cluster Former and Locality Sensitive Hashing is marginal -- less than one point on all measures. I am not sure the results are strong enough to support that clustering is better than random assignments. For a valid comparison, they should report the results with locality-sensitive hashing and 512 buckets.

The paper evaluates on three QA datasets, as long as experiments on perplexity for language modeling and shows promising performances.

Some clarifying questions:
1) could you specify a bit more on how do "classify the mean values of the first hidden state of all the chunked sequences to identify whether the question has short / long answers or not"?
2) I'm a bit confused with the experimental set up. For NQ, what's the numbers in Table 2? Is it on the dev set, and the numbers on Table 3 are on the test set? Please make it clear.
3) would this work on a really lengthy QA dataset such as narrativeQA?
4) From Table 2, it seems the more the number of clusters, the better the performance.  Why do you stop at 512? Is this have something to do with computational efficiency?

---

> ### Author Response · Authors · 2020-11-19
> **Response to Reviewer #3**
>
> Thank you for your insightful comments. We have updated the draft with the modifications in blue. Below, we provide detailed responses to your questions.
>
> Q1: Why would communicate between similar vectors more efficient than dissimilar or randomly chosen vectors? Would the performance improve if you use a better clustering algorithm?
>
> A1: According to LSH in Reformer (Kitaev et al., 2020), “the queries are sorted by bucket number”. In this way, if using randomly chosen vectors, one extreme case would be that most of the vectors are assigned to one or two buckets. In this case, sorting by bucket number will fail, as most vectors share a same number. By using K-Means, it automatically computes the bucket centroids which will follow the distribution of hidden states. It can make the model more robust and less likely to fail the sorting. Thus, we do agree that a better clustering algorithm will make the model better and more robust. While considering the speed of clustering, we mainly rely on K-Means from Nvidia.
>
> Q2: Experiments of LSH with more buckets.
>
> A2: The results of LSH with 512 buckets are also worse than our method after tuning the dropout rate. The following is about LSH best results on Quasar-T, SearchQA and NQ respectively: 53.4/63.5, 67.2/74.6, 76.0/56.7, compared to our best results 54.0/63.9, 68.0/75.1, 76.5/57.1.
>
> Q3: How to "classify the mean values of the first hidden state of all the chunked sequences to identify whether the question has short / long answers or not".
>
> A3: For Natural Question, we first identify whether the question has short/long answers or not based on the mean values of the first hidden state of all the chunked sequences, $\frac{1}{K}\sum_{k=1}^{K}\mathbf{H}^{N}_k[0]$ , where $K$ is the number of chunks and $N$ is the number of layers. We have modified the description in the updated draft.
>
> Q4: The experimental set up of NQ on Table 2 and 3.
>
> A4: All the results of NQ (including baselines) on Table 2 are based on dev set and Table 3 based on test set. We have modified the captions of the tables in the updated draft.
>
> Q5: Would this work on a really lengthy QA dataset such as narrativeQA?
>
> A5: The problem of NarrativeQA is quite close to OpenQA setting (like Quarsar-T and SearchQA). If we can retrieve the question related context from the book by an information retriever, our model (a reader) is able to extract the answer from the context. In this work, we mainly focus on improving the reader part instead of information retriever, similar to BigBird and ETC works.
>
> Q6: From Table 2, it seems the more the number of clusters, the better the performance. Why do you stop at 512? Is this have something to do with computational efficiency?
>
> A6: We had experiments on 1024 clusters and it will not further boost the performance. As shown in Table 1, the median number of tokens in context for different datasets are 2.8k, 2.5k, 6.3k which are not too long. Too many clusters cannot significantly boost performance.

---

### Official Review · AnonReviewer1 · 2020-10-29
**Good results, but not a new work**

**Rating:** 2
**Confidence:** 5

**Review:**

The paper proposes ClusterFormer to address the problem of quadratic compute requirements of the attention mechanism in a Transformer model. To this end this paper proposes to combine local attention to promote local consistency and proposes KMeans clustering to gather global information for every token. The paper establishes strong results on the long form question answering task of Natural Questions in an extractive setup, with it getting the leaderboard position ahead of ETC-large. While the idea in the paper is natural and the results on NQ are strong, unfortunately the idea in the paper is not new and has already been introduced in the work "Efficient Content-based Sparse Attention with Routing Transformers" [1, 2] which the authors fail to cite or credit. Therefore, I recommend rejection.


References:

[1] https://openreview.net/forum?id=B1gjs6EtDr

[2] https://arxiv.org/abs/2003.05997

---

> ### Author Response · Authors · 2020-11-19
> **Response to Reviewer #1**
>
> Thank you for your insightful comments.  We are sorry that we missed to cite Routing Transformer, which has been added in the revision. Below, we provide detailed comparison to Routing Transformer.
>
> Q: Comparison to Routing Transformer
>
> A: Although Routing Transformer also uses clustering-based method to build sparse attention, we are different regarding the following aspects:
>
> 1) Frameworks are different. Routing Transformer is based on the combination of local attention and routing attention, and focuses on Auto-regressive Sequence Modeling tasks. Our Cluster-Former combines sliding window and cluster-former layers, and we focus on Question Answering tasks. Our framework needs to take question into consideration whenever encoding sequences in sliding windows.
>
> 2) Initializations are different. Routing Transformer is trained from scratch and not clear how well it works by initializing with pre-trained models, especially the limitation of 512 position embeddings from pre-trained models, such as BERT, RoBERTa, ALBERT. We use sliding window layers to overcome this issue, and our Cluster-Former can be easily initialized by pretrained models. All of our experiments are based on RoBERTa.
>
> 3) Cluster centroids are updated in different ways.  Routing Transformer updates each cluster centroid µ by an exponentially moving average of all the keys and queries assigned to it, $\mu \leftarrow \lambda \mu +(1-\lambda)Q/2+(1-\lambda)K/2$. Our Cluster-Former maintains a memory queue to save hidden states which are used to update cluster centroids by running KMeans periodically.
>
> As Routing Transformer and Cluster-Former work on different tasks, we haven’t found an easy way to run Routing Transformer on QA for fair comparison. We have added analysis of Routing Transformer in the updated draft and will try to re-implement Routing Transformer for comparison in the future work.
>
> Please do let us know if you think the above comparison is not reasonable! Thanks a lot!

---

> > ### Comment · AnonReviewer1 · 2020-11-19
> > **Sliding window and local attention are not different**
> >
> > Thanks for the response! Let me respond to the points you raised:
> >
> > 1. Frameworks are different. How is sliding window attention different from local attention? Local attention where every token attends to a window of k previous tokens (for the auto-regressive case) is equivalent to sliding window attention with window size k and stride 1. So there isn't any difference between local attention and sliding window attention. Regarding the task, yes, routing transformer focuses on auto-regressive tasks, while clusterformer focuses on QA, but that doesn't mean the framework is new.
> >
> > 2. You can initialize the local attention of routing transformer the same way, if your local attention window is 512 you can use the weights from a pre-trained BERT to initialize those. It is pretty much the same way the sliding window attention is initialized in Clusterformer.
> >
> > 3. The update rule may be different (although it is k-means anyway), it is not entirely clear how significant that is in the end performance.
> >
> > Thanks for adding the comparisons. I think it is reasonable. However, if the end motivation for ClusterFormer is to do long context understanding (which is what is stated in the Introduction), I would also encourage the authors to consider more tasks instead of just extractive QA datasets like NQ. One way to test how good it captures long context would be to try challenging generative datasets such as Imagenet-64 or PG-19 or long-form generative QA like ELI5.

---

> > > ### Author Response · Authors · 2020-11-21
> > > **Response to Reviewer #1**
> > >
> > > Thanks a lot for your quick reply and insightful comments!
> > >
> > > Q1: About different frameworks.
> > >
> > > A1: The sliding-window-based method is local attention. But we cannot set stride to 1 as local attention in Routing Transformer, as we are focusing on Question Answering tasks here but not language modeling. We make all context tokens have an equal chance to meet a question. Each window also needs to include the question. Stride 1 will take much more memory. It is not trivial to directly adopt Routing Transformer codebase to QA tasks. And we are not claiming the difference between local attention and sliding window. What we mean by saying different frameworks is that Routing Transformer is on different head-level attention mechanisms, while Cluster-Former is on layer level. As we think it is particularly important to get local information in the bottom layers, our cluster-former layer only applies to deeper layers. According to our experiments, adopting cluster-former in shallow layers (first or second layers) will introduce noise and lead to poorer performance, as we only have 512 position embeddings from pretrained models to encode 5K tokens.
> > >
> > > Q2: About initialization and update rule
> > >
> > > A2: Yes, we agree that only experiments can show how Routing Transformer works. But Routing Transformer is not explicitly designed for QA. It does not use any pre-trained model. And our centroids update rules are quite different in two aspects regarding kmeans: 1) for the centroid assignment step, Routing Transformer works on the representations in a single batch, while ClusterFormer works on a queue of 100K representations in memory; 2) for the centroid update step, Routing Transformer is momentum update batch by batch, while ClusterFormer is periodically updated based on 100K representations. It is not trivial to ship Routing Transformer to QA. Actually, we haven't found any open-sourced baselines that are easy to use for our task, and we have re-implemented two popular baselines (Sparse Transformer and Reformer).  We have beat several SOTA models on NQ. We will try to add a comparison with Routing Transformer in the future.
> > >
> > > Q3: Experiments on more tasks
> > >
> > > A3: Thank you for your suggestions! We will fix our motivation soon: our Cluster-Former is specifically designed for extraction-based question answering. For experiments on challenging generative datasets, we will search for codebases that can replicate SOTA on these tasks, and try our methods in the future.
> > >
> > > Please let us know if you have any additional questions, and we are happy to further address them. Thank you.

---

> > > > ### Comment · AnonReviewer1 · 2020-11-21
> > > > **Differences with Routing Transformer are minor**
> > > >
> > > > Thanks for your response.
> > > >
> > > > 1. Right, so since sliding window attention and local attention are indeed one and the same thing in terms of architecture, the size of the window and the strides one chooses for the window are hyper-parameter choices which one can change or tune for a specific problem. A Transformer model with 16 layers is still called a Transformer even though the original hyper-parameter from the paper [1] had 12 layers. Regarding your point about ClusterFormer employing clustering on the layer level instead of the head level - again that is a hyper-parameter choice.
> > > >
> > > > "As we think it is particularly important to get local information in the bottom layers, our cluster-former layer only applies to deeper layers. "
> > > >
> > > > Again, this is something that has also been pointed out in multiple works including the Routing Transformer work [2, 3] and is not a new observation. Indeed the public implementations of Routing Transformer [4, 5] have specific hyper-parameters for skipping clustering in the lower layers - e.g., the hyper-parameter is called "sparsity_skip_first" in [4].
> > > >
> > > > 2. Again see point 1. The Transformer paper had results only on Machine Translation, however if we now use a Transformer model for language modeling or Question Answering or summarization it is still called a Transformer model, just the task is different. Regarding pre-trained models there is a pre-trained left to right (LM) model in [4]. But it really depends on your task - you can train a Routing Transformer on the same data-set that you pre-trained on for comparison.
> > > >
> > > > Again, I acknowledge that the training specifics of your mini-batch k-means may be slightly different. This then becomes an issue of the learning algorithm, e.g., do you use SGD or Momentum or Adam etc. It is not clear how the different learning algorithms impact the end performance on the QA task.
> > > >
> > > > I also acknowledge that you get SOTA on long form QA, but that is not really a popular long sequence benchmark, see for example [6] for PG-19 and Imagenet-64. Moreover, just getting SOTA on a benchmark is not a sufficient condition for acceptance of a paper - getting SOTA means that the architecture being proposed is good. But is the architecture new/novel? If the architecture builds off a closely related architecture such as Routing Transformer in this case, then one must argue that the difference (which is really the k-means learning algorithm) helps in some way over the original architecture.
> > > >
> > > > 3. Thanks, yes a closer ablation of the difference in the k-means learning algorithm would be more helpful in this regard than getting SOTA on some leaderboard. For example for optimizing neural nets there is some motivation for using Momentum/Adam over SGD. Can one have some similar motivation for the learning algorithm you use? This will become clear when you compare your method on benchmarks in common with Routing Transformer.
> > > >
> > > > References:
> > > >
> > > > [1] https://arxiv.org/abs/1706.03762
> > > >
> > > > [2] https://arxiv.org/abs/2003.05997
> > > >
> > > > [3] https://www.aclweb.org/anthology/2020.acl-main.672/
> > > >
> > > > [4] https://github.com/google-research/google-research/tree/master/routing_transformer
> > > >
> > > > [5] https://github.com/lucidrains/routing-transformer
> > > >
> > > > [6] https://deepmind.com/blog/article/A_new_model_and_dataset_for_long-range_memory

---

> > > > > ### Author Response · Authors · 2020-11-24
> > > > > **Response to Reviewer #1**
> > > > >
> > > > > Thank you for your quick feedback and detailed comments again!
> > > > >
> > > > > Q1: About differences between Routing Transformer and our Cluster-Former
> > > > >
> > > > > A1: Please let us summarize the differences:
> > > > >
> > > > > 1) One significant difference is that Routing Transformer never considers questions. It is only evaluated on Auto-regressive Sequence Modeling tasks. We are focusing on different tasks. More fair comparison on QA tasks should be with BigBird[1], ETC[2], etc., as we have done.
> > > > >
> > > > > 2) Our cluster centroids are updated in different ways: Routing Transformer online updates and ClusterFormer offline updates.
> > > > >
> > > > > 3) Based on the previous difference, we do layer-wise clustering and Routing Transformer head-wise clustering. Our method is more efficient as we only cluster once per layer, and we don’t need to merge both key and query vectors for clustering.
> > > > >
> > > > > 4) The methods of cluster assignment are different. Routing Transformer selects top-k input vectors for each cluster. Same vectors can be assigned to different clusters. However, Cluster-Former is more like a process of first sorting and then chunking. Note that we assign each input vector to its closest centroid, and we greedily sort out the cluster centroids, so that the vectors assigned to closer centroid ids are also close to each other. After sorting the cluster id of all the input vectors, we can do the chunking.
> > > > >
> > > > > 5) Our initializations are different. Our model is initialized by RoBERTa and we have proved it works. For Routing Transformer, it is still unclear now, although it is also possible.
> > > > >
> > > > > Q2: About long-form QA
> > > > >
> > > > > A2: We think long-former QA is a popular task, and we would like to focus more on the NLP side other than the machine learning side. PG-19 and Imagenet-64 are to test the ability to model long dependency, but not real applications. Both Big Bird[1] and ETC[2] don’t have experiments on PG-19 and Imagenet-64, but focus on real problems, such as QA. They are also very popular works compared to other efficient Transformers. In this paper, as stated in our title, our method is specifically designed for QA, and we achieved SOTA on three QA benchmarks.
> > > > >
> > > > > Q3: About fair comparison to Routing Transformer
> > > > >
> > > > > A3: As we described in A1, there are many differences between Routing Transformer and Cluster-Former. We must tune a lot to make Routing Transformer work for QA. We will make a fair comparison in the revision.
> > > > >
> > > > > [1] (NeurIPS 2020) Big Bird: Transformers for Longer Sequences,
> > > > >
> > > > > Manzil Zaheer, Guru Guruganesh, Avinava Dubey, Joshua Ainslie, Chris Alberti, Santiago Ontanon, Philip Pham, Anirudh Ravula, Qifan Wang, Li Yang, Amr Ahmed
> > > > >
> > > > > [2] (EMNLP 2020) ETC: Encoding Long and Structured Inputs in Transformers,
> > > > >
> > > > > Joshua Ainslie, Santiago Ontanon, Chris Alberti, Vaclav Cvicek,  Zachary Fisher, Philip Pham, Anirudh Ravula, Sumit Sanghai, Qifan Wang, Li Yang

---

### Official Review · AnonReviewer4 · 2020-10-29
**Effective attention for long sequences via chunking and clustering.**

**Rating:** 6
**Confidence:** 4

**Review:**

Summary:
Cluster-former is the latest proposal for enabling transformers to deal with long input sequences. Such sequences are particularly problematic for problems like question answering, QA, (or summarization), where the context can be arbitrarily long, and effectively open-ended when the setup includes a context retrieval component (e.g., as in OpenQA). Cluster-Former combines local information by encoding sequence chunks separately with a sliding window, then injects clustering layers, that use k-means to compute centroids to cluster hidden states and capture global information. The approach yields state-of-the-art, and top-of-leaderboard, results on Natural Questions (long answers).

This is great solid work, showing how clustering can be designed, implemented and  used successfully, to capture long distance dependencies in sparsified self-attention models. This is a concrete and useful contribution in itself for the large community working on this type of architecture and related problems. At the same time the approach involves quite a bit of complexity which makes one wonder if the baselines could be more competitive given a comparable amount of fine tuning. At the same time, competitive solutions of different nature (generative) are being proposed that pose a concrete challenge to this type of architecture, which are not evaluated, but should be at least discussed.

Pros
- Solid proof of concept and reference to successfully implementing clustering in sparse attention.
- Strong empirical results, particularly the Natural Questions’ leaderboard for long answers.
- Impressive amount of technical work, also with respect to reproducing results with other systems.
- Notwithstanding the amount of work in this area, literature review and comparison seems adequate but I might have missed something.
- Some qualitative analysis: which could be extended and articulated, in particular it would be interesting to understand where the long distance information helps; e.g., vs the sliding window approach and particularly vs LSH.

Cons
- One of the arguments for the paper is that it is not clear if related methods, like Reformer, can generalize to long sequences. However, in the evaluated implementation (Table 2) LSH is not that much worse than k-means. In fact, even just the sliding window alone seems surprisingly competitive on all QA tasks. While being much simpler. I find the authors’ effort to compare with all these related methods truly commendable. It seems natural to wonder how much more fine-tuning has gone into Cluster-Former compared to the simpler baselines, given its additional complexity. It would be important to discuss this aspect in more detail.
- Given the recent work of generative readers: https://arxiv.org/abs/2005.11401, and particularly Izacard & Grave, (FID, https://arxiv.org/pdf/2007.01282.pdf) it seems unclear that direct encoding is the only, or the best, option for dealing with long sequences, at least for QA. In particular, FID seems attractive due to its simplicity and capacity (about twice as much as Cluster-Former it seems). The authors should discuss this work. It would be ideal, at some point, to compare directly by evaluating on OpenQA-NQ or by other means.

Detailed feedback
- Pleas define x, from x\times d, right below Eq(1). Num tokens in context?
- Scaler value/scalar value?
- It would be great to explain Eq(2) step by step for clarity.
- What is the effect of the overlapping content size m-l? And in general of parameters l and m. In particular, could this affect positively the performance of the simpler sliding window model?
- Why using cluster layers at only 2 fixed depths? How does this parameter affect results?
- The max length is constrained to 5k (10k test) due to memory constraints, can this be improved, how?
- How long did it take to train the leaderboard (NQ) entry system?
- Unclear what table 2 evaluates on, e.g., for NQ, is this on the dev set? Or a fraction of it?

---

> ### Author Response · Authors · 2020-11-19
> **Response to Reviewer #4**
>
> Thank you for your encouraging and insightful comments. We have updated the draft with the modifications in blue. Below, we provide detailed responses to your questions.
>
> Q1: About comparison to baselines such as LSH, and how much more fine-tuning has gone into Cluster-Former compared to the simpler baselines.
>
> A1: We really appreciate your acknowledgment of our efforts on baselines. Most of the baselines are either not open-sourced or cannot be easily transferred to our tasks. To make a fair comparison, we re-implement LSH and Sparse Attention, and integrate them into our framework. We fix the sliding window layers for all the methods. To select which layers should be used for sliding windows, we only have a wide exploration of it on Quasar-T dataset based on Cluster-Former. Then we fix sliding windows layers for the other datasets, and only tune dropout from {0.1, 0.15, 0.2} for all the methods including baselines and report the best result. We have updated it in the draft.
>
> Q2: Comparison to generative readers and FID
>
> A2: Yes, we agree that generative reader is a good solution to overcome reading long sequences. However, it still depends on how well the retrievers are. As pointed out by FID, “we show that the performance of our method significantly improves when the number of retrieved passages increases”, it is still important to have a better reader to read longer context. The main reason we didn’t test our models on OpenQA-NQ is that the retrieved passages are not the same in different methods. Thus, to make a fair comparison with previous readers, we focus more on the datasets with fixed context. We will try to evaluate our model on OpenQA-NQ during the discussion or in the final version.
>
> Q3: Pleas define x, from x\times d, right below Eq(1). Num tokens in context? Scaler value/scalar value? It would be great to explain Eq(2) step by step for clarity.
>
> A3: Thank you for your suggestions! We have modified them and rewritten the explanation for Eq(2) step by step in the updated draft.
>
> Q4: What is the effect of the overlapping content size l-m?
>
> A4: We had experiments on (l=256, m=224) and (l=256, m=230) for sliding-window-only baseline on Quasar-T. There is no significant difference, and we select (l=256, m=224) for all our experiments.
>
> Q5: Why using cluster layers at only 2 fixed depths? How does this parameter affect results?
>
> A5: In Table 4, we have a hyper-parameter search for using cluster layers. It also includes experiments with 3/4/5/6 fixed depths. And we select the best hyper-parameter for all the other datasets.
>
> Q6: The max length is constrained to 5k (10k test) due to memory constraints, can this be improved, how?
>
> A6: Yes, as our method doesn’t have the quadratic issue on sequence length, one solution is to map 24 layers to multi-GPUs, and another solution is to call checkpoint function, such as “torch.utils.checkpoint.checkpoint” from Pytorch, which does not save intermediate activations, and instead recomputes them in backward pass. However, both methods will make the encoding slower and we will try it to encode longer sequences in future work.
>
> Q7: How long did it take to train the leaderboard (NQ) entry system?
>
> A7: It takes one day by using 8 V100 GPU.
>
> Q8: Unclear what table 2 evaluates on, e.g., for NQ, is this on the dev set? Or a fraction of it?
>
> A8: For NQ, all the results including baselines are on the full dev set. And the other datasets are on the test set.

---

### Decision · Program_Chairs · 2021-01-07
**Final Decision**

**Decision:**

Reject

**Comment:**

The paper attempts to make transformers more scalable for longer sequences. In this regards, authors propose a clustering-based attention mechanism, where only tokens attends to other tokens in the same cluster. This design reduces memory requirements and allows more information mixing than simple local windows. Using the proposed approach, new state-of-the-art performance is obtained on Natural Questions long answer, although marginal. However, reviewers raised numerous concerns. First, the novelty of the paper compared to prior work like reformer or routing transformer which also conceptually does clustering is not resolved. Second, the claim that k-means yields a more balanced/stable clustering than LSH is not well established. Finally, why clustering, i.e. attention between similar vectors is better than dissimilar or randomly chosen vectors or does is it even as expressive is not clear. Thus, unfortunately I cannot recommend an acceptance of the paper in its current form to ICLR.